# A New Cationic Fluorescent Probe for HSO_3_^−^ Based on Bisulfite Induced Aggregation Self-Assembly

**DOI:** 10.3390/molecules27082378

**Published:** 2022-04-07

**Authors:** Xing Zhang, Shao-Yuan Su, Xuan-Ting Chen, Ling-Yi Shen, Qi-Long Zhang, Xin-Long Ni, Hong Xu, Zhi-Yong Wang, Carl Redshaw

**Affiliations:** 1The Key Laboratory of Environmental Pollution Monitoring and Disease Control, Ministry of Education, School of Public Health, Guizhou Medical University, Guiyang 550004, China; zhangxing11207115@126.com (X.Z.); chenxuanting345@163.com (X.-T.C.); shenly@stumail.nwu.edu.cn (L.-Y.S.); xuhong@gmc.edu.cn (H.X.); 2Key Laboratory of Macrocyclic and Supramolecular Chemistry of Guizhou Province, Guizhou University, Guiyang 550025, China; phw2021@126.com; 3Department of Chemistry, University of Hull, Cottingham Road, Hull HU6 7RX, UK; c.redshaw@hull.ac.uk

**Keywords:** fluorescent probe, aggregation self-assembly, HSO_3_^−^ recognition

## Abstract

In comparison with the numerous studies that have centered on developing molecular frameworks for the functionalization of fluorescent materials, less research has addressed the influence of the side chains, despite such appendages contributing significantly to the properties and applications of fluorescent materials. In this work, a new series of cationic fluorescent probes with AIE characteristics have been developed, which exhibit unique sensitivity for charge-diffusion anions, namely HSO_3_^−^, via the interactions of ions and the cooperation of the controllable hydrophobicity. The impact of the alkyl chain length attached at the cationic probes suggested that the fluorescent intensity and sensitivity of the probes could be partially enhanced by adjusting their aggregation tendency through the action of the hydrophobic effect under aqueous conditions. DLS and SEM images indicated that different particle sizes and new morphologies of the probes were formed in the anion-recognition-triggered self-assembly process, which could be attributed to the composite effect of electrostatic actions, Van der Waals forces and π-π stacking.

## 1. Introduction

Sulfur dioxide (SO_2_) is a common air pollutant, which can easily be converted into SO_2_ derivatives (HSO_3_^−^/SO_3_^2−^) and can thereby lead to multiple respiratory diseases, such as rhinitis, chronic bronchitis and chronic obstructive pulmonary disease (COPD) [1]. HSO_3_^−^ has been considered as a promising biomarker to monitor the metabolic transformation process in biological systems, and is a metabolite of some sulfur-containing amino acids [2]. In the food industry, despite sulfite and bisulfite derivatives being utilized as antioxidants to prevent food from oxidative deterioration and for extending shelf life, the excessive amounts of HSO_3_^−^/SO_3_^2−^employed in food processing is harmful to human health, such that any increased intake can lead to various disorders such as allergic reactions, asthma and nervous system disorders [3,4]. Thus, the development of a new convenient and efficient detection method for HSO_3_^−^ will be a useful aid for body health.

Organic fluorescent probes have attracted plenty of attention in recent years because of their stable structures, high detection efficiency and non-invasive features in comparison to other sensors, and so have been utilized in numerous fields [5,6,7,8]. A diverse range of fluorescent probes have been designed according to the principles of Photoinduced Electron Transfer (PET), Intramolecular Charge Transfer (ICT), Excited State Intramolecular Proton Transfer (ESIPT), Fluorescence Resonance Energy Transfer (FRET), Excimer/Exciplex and Aggregation-Induced Emission (AIE) [9]. Among them, probes with AIE characteristics have received much interest since 2001, and the property of aggregation in solvents such as water or aqueous-containing conditions, which results in bright emission, suggests they have great potential for use in biomaterials [10,11,12,13]. With this in mind, a highly selective sensor that could be applied in high water content environments would be a promising tool for sulfate derivative detection.

In this research, a series of new fluorescent probes for the detection of HSO_3_^−^ based on a 2,3,5,6-tetramethylbenzene core were designed by employing simple synthetic routes. The probe TBPD exhibited weak emission at 505 nm in the free or non-aggregation state. In addition to HSO_3_^−^, the phenomenon of aggregation self-assembly of the TBPD induced by HSO_3_^−^ in aqueous conditions was observed, which led to the appearance of a new emission peak at 535 nm with a bright buff light. In addition, the sensitivity of the TBPD could be improved by adjusting the length of the alkyl chain attached to the pyridine ring; the detection limit was calculated to be as low as 0.5 μM.

## 2. Results and Discussion

The gemini-surfactant-like sensors **TBPD^2+^-10C, TBPD^2+^-11C,** and **TBPD^2+^-12C** were synthesized from a fixed central aromatic core, to which two aliphatic chains were attached at the pyridinium rings from a two-step reaction (Figure 1). These series of probes exhibited several interesting properties given by the special A-D-A structure: (1) the π-conjugated tetramethylbenzene core served as the main electron donor and fluorescent group. The pyridinium rings serve as the electron acceptor part with anion binding sites, which further increased the water solubility of the probes. The attached aliphatic chains also play an important role and contribute to the regulation of molecular solubility. (2) An exciting intramolecular charge transfer (ICT) process can occur from the tetramethylbenzene core to the pyridinium motif, [14] and this phenomenon is consistent with the result whereby the fluorescence wavelength was red-shifted on increasing the solvent polarity (Appendix A) [15]. In addition, the intramolecular C-C bond between the aromatic rings and the pyridine moieties can rotate freely within these molecules, which will reduce the radiative transition energy and the molecular planarity, [16] and this together results in a weak fluorophore or no emission. (3) It can be observed that with the increase in alkyl chain length, the emission wavelength of solvents gradually undergoes a red-shift (Appendix A), and this may be attributed to the hydrophobicity of the sensors changing upon modification of the aliphatic chain, and the strength of the electron transfer effect within the molecules donated by the aliphatic chain is also alterant.

The critical aggregation concentration (CAC) of the sensors in water was evaluated by fluorometry. As shown in Figure 1, the CACs of the series of sensors were determined to be 7.38 × 10^−6^ M for **TBPD^2+^-10 C**, 8.50 × 10^−6^ M for **TBPD^2+^-11 C** and 2.10 × 10^−5^ M for **TBPD^2+^-12 C**, respectively. Based on this data, we can conclude that the CACs of these probes decreased on increasing the length of the fatty chain, in other words, the length of the fatty chain played an important role in the process of hydrophobic self-assembly in aqueous conditions. The longer the alkyl chain, the more hydrophobic the probe will be, and so a larger particle size will form [17]. This proposal was also corroborated by the dynamic light scattering (DLS) experiments: the particle size of **TBPD^2+^-10 C**, **TBPD^2+^-11 C** and **TBPD^2+^-12 C** was recorded at 51 nm, 53 nm and 75 nm, respectively (Appendix A).

The powdered sensors exhibited a bright red emission at 692 nm (Appendix A), but exhibited only very weak yellow light in dilute solution. It is proposed that the emission behavior may be affected by the AIE effect, thus, we selected THF/H_2_O as the test solvent system to further explore the behavior. As shown in Figure 2, **TBPD** exhibited weak emission in pure THF solution, and as the water fraction (*f_w_*) increased from 0% to 70%, the fluorescent intensity (FL) of the probes barely changed. However, when the water content reached 80%, some tiny floating particles could be observed in the solution, and the FL of the solvents soared dramatically. In pure water solution, the FL of the probe attained the maximum value with an approximately 3.5-fold increase vs. that in the pure THF solution, which is consistent with AIE characteristics for **TBPD** [18,19].

Next, an ion recognition experiment was carried out via measurement of the fluorescence response of the sensors (5.0 μM) in a neutral water-containing medium (*f_w_* = 70%) upon the addition of 100.0 equiv. of various anionic ions (such as Cl^−^, SO_4_^2−^, SO_3_^2−^, ClO_4_^−^, Ac^−^, NO_3_^−^, HSO_3_^−^, PF_6_^−^, CO_3_^2−^, F^−^, I^−^, BF_4_^−^, H_2_PO_4_^−^ and SCN^−^).

As shown in Figure 3, upon the addition of various anions to the mixture, only HSO_3_^−^ induced a significant change of the emission behavior for **TBPD^2+^-10C**, **TBPD^2+^-11C** and **TBPD^2+^-12C**. The fluorescence spectra of the solvents exhibit a new similar monomer emission wavelength with a maximum emission peak located at 542 nm for **TBPD^2+^-10C**, 539 nm for **TBPD^2+^-11C** and 537 nm for **TBPD^2+^-12C**, while other anions had only a limited effect on their fluorescent behavior, besides, under a 365 nm UV lamp irradiation, the fluorescent color of the solvents changed from dark to yellow light.

Compared to the free probe solution, the fluorescent intensity of the solvents containing HSO_3_^−^ induced an approx. 7-fold enhancement for **TBPD^2+^-10C**, a 28-fold enhancement for **TBPD^2+^-11C** and a 52-fold increase for **TBPD^2+^-12C** (Appendix A). Competitive experiments were then performed to explore the selective recognition ability for the series of sensors toward HSO_3_^−^. Upon the addition of 100 equiv. of coexisting ions (such as Na^+^, K^+^, Li^+^, Pb^2+^, Ca^2+^, Mg^2+^, Cu^2+^, Zn^2+^, Cd^2+^, Co^2+^, Ni^2+^, Cl^−^, SO_4_^2−^, SO_3_^2−^, ClO_4_^−^, Ac^−^, NO_3_^−^, PF_6_^−^, CO_3_^2−^, F^−^, I^−^, BF_4_^−^, H_2_PO_4_^−^ and SCN^−^) to the mixture containing the probe and HSO_3_^−^, the emission behavior barely changed (Appendix A), which indicated that the probes **TBPD^2+^-10C**, **TBPD^2+^-11C** and **TBPD^2+^-12C** exhibited good specificity and selectivity toward HSO_3_^−^.

Fluorescence titration experiment using these probes under aqueous conditions with different amounts of HSO_3_^−^ showed a linear relationship between the solvent emission intensity of their fluorescent peaks and the concentration of HSO_3_^−^ added (Appendix A). The limit of detection (LOD) of the probes for HSO_3_^−^ were calculated to be 4.30 μM (**TBPD^2+^-10C**), 0.72 μM (**TBPD^2+^-11C**) and 0.50 μM (**TBPD^2+^-12C**). The sensitivity of the sensors was enhanced by increasing the length of the fatty chain attached to the pyridine rings. Furthermore, the series of sensors exhibited good acid and alkali resistance capacity towards HSO_3_^−^ recognition (Appendix A).

To better understand the recognition mechanism of the series sensors, some essential experiments were performed. Firstly, we utilized methanol as the recognition medium to explore whether the polarity of solvents or H_2_O was favorable for the desolvation of the charge diffusion of HSO_3_^−^, and the results (Appendix A) suggested that no obvious changes had been observed after various anions were added into the mixture, which indicated that H_2_O may benefit the process of charge-diffuse anion recognition; by the means of the hydrophobic effect of these series of cationic sensors in water- containing condition, the different binding strength between the negatively charged HSO_3_^−^ and positively charged sensors enable the identified anions HSO_3_^−^ to suitably trigger the process of the sensor aggregation by the combinative effect of electrostatic ionic bonding, π-stacking of the π-conjugated aromatic moieties, and Van der Waals forces between the intermolecular alkyl chains [17,20,21,22].

Secondly, the dynamic light scattering (DLS) experiments were conducted to investigate the average particle size of the probe- HSO_3_^−^ complex under aqueous conditions. As shown in Figure 4, upon addition of HSO_3_^−^, the average particle size of **TBPD^2+^-10C**, **TBPD^2+^-11C** and **TBPD^2+^-12C** changed from 51 nm, 53 nm and 75 nm to 302 nm, 377 nm and 528.91 nm, respectively, which suggested that in the presence of HSO_3_^−^, the probe molecules formed a new “aggregation state” as the model of coexistence; in other words, the HSO_3_^−^ induced the ionic-interaction-induced aggregation self-assembly phenomenon. [23,24] As a consequence, the solution containing the **TBPD**s-HSO_3_^−^ complex exhibited brighter emission than the free probes under the impact of such a new aggregated form.

Finally, in order to obtain more detailed information about the self-assembly process induced by HSO_3_^−^, SEM tests were performed. Taking **TBPD^2+^-12C** as an example, as shown in Figure 5, the image of free **TBPD^2+^-12C** was a solid rod with a length of tens of microns. Whereas following the addition of HSO_3_^−^ to the mixture, the morphology changed into a few microns of loose rods stacked together. On the other hand, the results of energy dispersive X-ray spectroscopy (EDS) elemental analysis demonstrated that S and O were evenly distributed in the large rods, which suggested that the target ion HSO_3_^−^ did combine with the probes [25].

According to the experimental evidence mentioned above, we inferred the recognition mechanism of the sensors towards HSO_3_^−^ as follows (Figure 6): in the absence of HSO_3_^−^, based on the interaction of Van der Waals forces between the intermolecular fatty chains and the π-stacking of the π-conjugated aromatic moieties, the probe molecules adopted an alkyl chains-to-alkyl chain and an aromatic nucleus-to-aromatic nucleus mode, thereby affording the solid rod structure [26,27]. Meanwhile, with the addition of HSO_3_^−^, and under the action of electrostatic forces of positive and negative charges, the pyridine rings containing anion binding sites and the attached tetramethylbenzene cores formed “soybean pod”-like motifs, which had two “magnetic flagellas” at their ends, and subsequently, the “bean” HSO_3_^−^ was firmly embedded between two adjacent probe molecules. On the other hand, the sandwich formation also reduced the interaction of the π-π stacking between the intermolecular aromatic rings. In addition, due to the influence of the hydrophobic effect caused by H_2_O and Van der Waals forces between the intermolecular fatty chains, some of the stacking structure was broken and twisted, [14,17,28] and a shorter hollow nanotube structure was observed.

## 3. Experimental Section

### 3.1. Apparatus and Reagents

Unless otherwise stated, all the experimental materials and reagents were commercially available and used without further purification. The stock solutions of anions and cations were prepared from their sodium and nitrate salts, respectively. Ultraviolet-visible (UV-vis) absorption spectra were recorded on an Agilent 8453 mode spectrofluorometer (Agilent, Santa Clara, CA, USA). Fluorescence (FL) spectra were performed on an Agilent Cary Eclipse spectrofluorometer (Agilent, Santa Clara, CA, USA). ^1^H NMR and ^13^C NMR spectra were recorded at 400 MHz with an Inova-400 Bruker AV 400 spectrometer (Bruker, Karlsruhe, Germany) using Me_4_Si as the internal reference. High-resolution mass spectra (HRMS) were performed on a GCT premier CAB048 mass spectrometer (Waters Corp, Milford, MA, USA) operating in MALDI-TOF mode. The ESI-TOF spectra were recorded on an Agilent 6545 time-of-flight mass spectrometer (Agilent, Santa Clara, CA, USA). The particle size data were collected on a Malvern Zetasizer Nano-ZS light dispersion meter (Malvern Zetasizer, Shanghai, China). The scanning electron microscopy (SEM) was carried out using a Hitachi S-4800 II field-emission SEM system (Hitachi, Tokyo, Japan).

### 3.2. Synthesis and Characteristics of TBPD^2+^-10C, TBPD^2+^-11C and TBPD^2+^-12C

Synthetic route to TBPD: A mixture of 1,4-dibromo-2,3,5,6-tetramethylbenzene (584.0 mg, 2.0 mmol), 4-vinylpyridine (504.0 mg, 4.8 mmol), palladium acetate (22.5 mg, 0.1 mmol), triphenylphosphine (65.6 mg, 0.25 mmol) and K_2_CO_3_ (400 mg, 2.9 mmol) were dissolved in 5 mL DMF with 3 mL triethylamine, and the reaction mixture was refluxed for 12 h at 100 °C. After cooling to room temperature, the reaction was added to 100.0 mL distilled water and extracted with CH_2_Cl_2_ for three times, and the organic phase was collected, washed three times with saturated brine, and dried with anhydrous MgSO_4_ for 1 h. After that, the mixture was filtered, the solvent was removed under vacuum, and the residue was dissolved in EtOH. In addition to water, a yellow precipitate could be observed, which was collected and washed with water and petroleum ether several times. The target product **TBPD** could be obtained as yellow crystals on recrystallization from ethanol (307 mg, 45% yield).

^1^H NMR (400 MHz, CDCl_3_): δ 8.60–8.57 (m, 1 H), 7.41 (d, *J* = 16.5 Hz, 1 H), 7.37 (d, *J* = 6.1 Hz, 1 H), 6.40 (d, *J* = 16.6 Hz, 1 H), 2.32–2.21 (m, 4 H); ESI-TOF: *m*/*z* calculated for C_24_H_24_N_2_: 340.19; found: 341.20.

Synthetic route to probes: 1.0 mmol **TBPD** was weighed out and dissolved in 10 mL DMF in a 25 mL round bottom flask, then, 6.0 mmol 1-decyl iodide was added into the solution and reacted for 24 h at 90°C. After that, some red precipitate formed and subsequently was washed with acetone and petroleum ether several times. The probe **TBPD^2+^-10C** was acquired after vacuum drying (368 mg; 59% yield). Synthetic routes to **TBPD^2+^-11C** and **TBPD^2+^-12C** were similar to the above, but the 1-decyl iodide was changed to 1-undecyl iodide and 1-dodecyl iodide for **TBPD^2+^-11C** and **TBPD^2+^-12C**, respectively. Following this method, a series of new compounds were acquired.

**TBPD^2+^-10C**: ^1^H NMR (400 MHz, DMSO-*d*_6_): δ 8.97 (d, *J* = 6.7 Hz, 1 H), 8.30 (d, *J* = 6.8 Hz, 1 H), 8.13 (d, *J* = 16.6 Hz, 1 H), 6.77 (d, *J* = 16.5 Hz, 1 H), 4.48 (t, *J* = 7.3 Hz, 1 H), 2.35–2.08 (m, 4 H), 1.88 (s, 1 H), 1.43–1.05 (m, 10 H); ^13^C NMR (100 MHz, DMSO-*d*_6_): 152.81 (s), 145.01 (s), 142.22 (s), 136.39 (s), 132.48 (s), 130.29 (s), 124.63 (s), 60.39 (s), 31.81 (s), 31.14 (s), 29.21 (dd, J = 31.0, 15.8 Hz), 25.95 (s), 22.63 (s), 18.02 (s), 14.51 (s); ESI-TOF: *m*/*z* calculated for C_44_H_66_N_2_^2+^: 622.52; found: 623.52.

**TBPD^2+^-11C**: ^1^H NMR (400 MHz, DMSO-*d*_6_): δ 8.98 (d, *J* = 6.8 Hz, 1 H), 8.30 (d, *J* = 6.8 Hz, 1 H), 8.13 (d, *J* = 16.6 Hz, 1 H), 6.77 (d, *J* = 16.5 Hz, 1 H), 4.48 (t, *J* = 7.3 Hz, 1 H), 2.32–2.11 (m, 3 H), 1.87 (s, 1 H), 1.37–1.03 (m, 9 H), 1.00–0.57 (m, 2 H); ^13^C NMR (100 MHz, DMSO-*d*_6_): 152.93–152.77 (m), 145.00 (s), 142.24 (s), 136.40 (s), 132.47 (s), 130.28 (s), 124.63 (s), 60.42 (s), 31.81 (s), 31.12 (s), 31.12 (s), 29.23 (dd, J = 31.8, 21.7 Hz), 25.96 (s), 22.61 (s), 18.00 (s), 14.48 (s); ESI-TOF: *m*/*z* calculated for C_46_H_70_N_2_^2+^: 650.55; found: 651.52.

**TBPD^2+^-12C**: ^1^H NMR(400 MHz, DMSO-*d*_6_): δ 8.98 (s, 1 H), 8.30 (s, 1 H), 8.21–7.96 (m, 1 H), 6.77 (dd, *J* = 16.3, 7.5 Hz, 1 H), 4.49 (d, *J* = 6.5 Hz, 1 H), 2.21 (d, *J* = 7.5 Hz, 3 H), 1.89 (s, 1 H), 1.23 (d, *J* = 12.6 Hz, 11 H), 0.82 (d, *J* = 6.9 Hz, 101 H); ^13^C NMR(100MHz, DMSO-*d*_6_): 152.80 (s), 145.03 (s), 142.32–142.01 (m), 136.40 (s), 132.52 (s), 130.51–129.79 (m), 124.64 (s), 60.38 (s), 31.86 (s), 31.15 (s), 29.78–28.73 (m), 25.97 (s), 22.64 (s), 18.03 (s), 14.52 (s); ESI-TOF: *m*/*z* calculated for C_48_H_74_N_2_^2+^: 678.58; found: 679.58.

The detailed ^1^H NMR, ^13^C NMR and HRMS spectra diagrams of the **TBPD**, **TBPD^2+^-10C**, **TBPD^2+^-11C** and **TBPD^2+^-12C** are listed in the supporting information (Appendix A).

### 3.3. General Methods for Instrument Test

**NMR spectroscopy test.** NMR spectra were determined at room temperature by JEOL ECS 400 AVANCE III spectrometer. The solvents used were Chloroform-*_d_* and DMSO-*d*_6_. The internal standard of NMR spectra was tetramethylsilane (TMS). An appropriate amount of **TBPD**, **TBPD^2+^-10C**, **TBPD^2+^-11C** and **TBPD^2+^-12C** were accurately weighed and dissolved in 500.0 µL deuterated solution to prepare a 1.0 mM standard solution for the later NMR test.

**Time of flight mass spectrometry test.** A total of 1.0 mg of the samples were accurately weighed and then dissolved in a 50.0 mL volumetric flask with chromatographic grade acetonitrile, after that, 1.2 mL of the mentioned solution was taken and filtered by 0.22 μm PTFE microporous membrane and stocked in a liquid phase bottle standing for the qualitative determination.

**UV****-****vis spectra and FL spectra test.** Preparation for the probe stock solution: taking TBPD^2+^-10C as an example, 62.3 mg **TBPD^2+^-10C** was accurately weighed and then dissolved in THF to obtain 1.0 mM probe stock solution—50 mL. The preparation methods for **TBPD^2+^-11C** and **TBPD^2+^-12C** stock solutions were the same as above. Preparation for the anion/cation stock solution: appropriate amount of anion and cation salts were weighed and then dissolved in deionized water to acquire 0.1 M ion stock solution for a later test. Preparation for Tris-HCl buffer: 121.20 mg of trimethylol aminomethane was dissolved in 1.00 L of ultrapure water, and then its pH was adjusted to 7.00 with a 0.10 M HCl solution and a 0.10 M NaOH solution. The experimental condition for FL test: the excitation wavelength for **TBPD^2+^-10C**, **TBPD^2+^-11C** and **TBPD^2+^-12C** was 348 nm and the test wavelength range was 450–750 nm, test voltage was 680 V for **TBPD^2+^-10C**, 650 V for **TBPD^2+^-11C** and 620 V for **TBPD^2+^-12C**; the launching slit was 5/5 nm. Unless otherwise stated, all UV-vis spectra and FL spectra tests were carried out under a Tris-HCl buffer contained condition, and the pH values of the test systems were adjusted to be 7.00.

**Solid fluorescence test.** A total of 5.0 mg of sample powder was weighed and added on a quartz slide at room temperature, and the supporting bracket was tilted for 45°. Test voltage: 650 V, slit: 5/5 nm, excitation wavelength: 342 nm and wavelength range: 660–740 nm.

**SEM test.** A moderate amount of the probe stock solution was taken and diluted into 5.0 μM test solution, then, 50-fold HSO_3_^−^ was added into the test solution. After standing for a few minutes, the mixture with HSO_3_^−^ and without HSO_3_^−^ were placed on a monocrystalline silicon wafer and naturally air-dried for a later SEM test.

**Dynamic light scattering (DLS) test.** After filtering by a 0.22 μm microporous membrane, the probe stock solution was diluted into 5.0 μM test solution and then an appropriate amount of HSO_3_^−^ was added to it, shaken well and pouring the mixture into a dynamic light sample bottle for the DLS test. All experimental data were obtained with a controlling of the standard error below 5%.

## 4. Conclusions

In summary, a series of alkyl-chain-appended cationic fluorescent probes with AIE properties were designed and applied in anion recognition by the action of ionic interactions involving the subtle cooperation of the hydrophobic effect under aqueous conditions. On increasing the alkyl chain length, the fluorescent intensity of the **TBPD** type sensors and the sensitivity towards HSO_3_^−^ was significantly improved. This detection process exhibited a novel recognition signal output for target ions via the effect of aggregation self-assembly assisted by various non-covalent interactions, such as electrostatic interactions, Van der Waals forces and π-π stacking. To the best of our knowledge, this is the first instance of an HSO_3_^−^ fluorescent sensor based on ionic-interaction-induced aggregation self-assembly. We believe this work will provide an efficient strategy to address the challenge of trace anion pollutant detection in aqueous conditions and biosystems, and opens up a new avenue for the design of side-chain manipulated sensors.

## Data Availability

No new data were created or analyzed in this study. Data sharing is not applicable to this article.

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
