# Peer review of "A New Cationic Fluorescent Probe for HSO3 Based on Bisulfite Induced Aggregation Self-Assembly"

_molecules, 2022, doi:10.3390/molecules27082378_

Round 1

Reviewer 1 Report

The article by Zhang et al is devoted to a fluorescent sensor for the determination of sulfite anions.
The work was done at a high level, well written, the conclusions are substantiated by the results of experiments. The proposed mechanism for the occurrence of a sensory response seems likely.
Small remarks:
1) In the synthesis scheme (Scheme 1), ethylazide is indicated as one of the reagents. This does not correspond to the synthesis procedure described in the experimental part.
2) It is not entirely clear whether there is a dependence of the response on the pH of the solution. The existence of sulfite and hydrosulfite ions depends on this parameter.
3) In some places there are problems with subscripts and superscripts in formulas, for example, page 11, listing materials in SI
The article may be accepted for publication.

Author Response

Dear reviewer:

       Thank you for your comments, we have revised our manuscript accordingly, the revised manuscript is attached in the enclosure.

The article by Zhang et al is devoted to a fluorescent sensor for the determination of sulfite anions.
The work was done at a high level, well written, the conclusions are substantiated by the results of experiments. The proposed mechanism for the occurrence of a sensory response seems likely.
Small remarks:
1) In the synthesis scheme (Scheme 1), ethylazide is indicated as one of the reagents. This does not correspond to the synthesis procedure described in the experimental part.

Response: Thank you for your comment, the reagent used for this reaction is triethylamine, we have revised this mistake in our manuscript.
2) It is not entirely clear whether there is a dependence of the response on the pH of the solution. The existence of sulfite and hydrosulfite ions depends on this parameter.

Response: Thank you for your comment, unless otherwise stated, all UV-vis spectra and FL spectra tests were carried out under a Tris-HCl buffer contained condition, and the pH values of the test systems were adjusted to be 7.00, we now state these experimental conditions in the related captions and in the experimental sections.(marked in blue)
3) In some places there are problems with subscripts and superscripts in formulas, for example, page 11, listing materials in SI.

Response: Thank you for your comment, we have revised this mistake in our manuscript.
The article may be accepted for publication.

Reviewer 2 Report

This article reports the preparation of a series of cationic fluorescent probes with variable alkyl chain length. This work studies the effect of chain length to their fluorescent intensity and sensitivity towards HSO3- anion. The type of aggregation-induced emission probes reported here have recently gained attention for applications in sensing and bioimaging.

The authors thoroughly study the binding behaviour of the three fluorescent probes obtained and its aggregation induced emission. Regarding the mechanism for the enhance in emission in the presence of HSO3- the authors provide some evidence based on SEM images, however this is only marginal evidence and it should be mentioned in the main text that the mechanism is assumed. This is an interesting study, and the manuscript is well written, and presented to high standards. This reviewer believes that the readers of Molecules will enjoy the content of the manuscript and considers it suitable for publication after the following minor points have been addressed:

  1. Scheme 1 – The conditions given for obtaining compounds TBPD-10C, TBDP-11C, etc… only mention DMF and 1-Decyl iodide. It should be specified that these conditions only produce TBDP-10C and add the conditions for obtaining TBDP-11C and TBDP-12C.
  2. Page 3 – Please report error ranges for the DLS particle sizes. Actually, looking at the DLS distributions for TBPD-10C and TBDP-11C (Figure S13) they are very similar .
  3. Page 4 final paragraph – Please report the counterions being used to introduce the anionic ions. Do they all share the same cation? Is it possible that there is an influence of the cation?
  4. During the anion binding experiments, a large amount of ions (up to a 100 eq. in some cases) are added to aqueous solutions. Have the authors looked at buffered solutions? Some anions will be more basic than others and the pH and ionic strength of the solutions can have an effect on the aggregation process that the authors might not have considered.
  5. Page 6 – Authors report a DLS measurement as 528.91 nm. This level of accuracy is unrealistic with this technique. Please report a value that takes into account the experimental error of the technique.
  6. Figure 5 – The label for the SEM image in b) is blurry and does not allow to visualize correctly the size reference for the image. Please improve the quality of the text in here to make it easier to read.
  7. Regarding the aggregation induced mechanism, it would have been interesting to see a thorough pH study of the dependence of fluorescence with pH.
  8. Figure S15 – The comparison of the fluorescence spectra does not specify the chain length of TBDP probe (black line). The figure caption does not specify either the number of equivalents of HSO3- for each sample.
  9. Figure S16 – Please indicate the magnitude and units for the Y axis in the plots presented.
  10. Figure S19 – It is not clear what the authors intend to show with this figure. The authors need to update the caption to explain better what the figure means. Currently it says: “Fluorescent intensity …. Within different pH recognition conditions” What are those conditions? What is the different pH?

Author Response

Dear reviewer:

       Thank you for your comments, we have revised our manuscript accordingly, the revised manuscript is attached in the enclosure.

This article reports the preparation of a series of cationic fluorescent probes with variable alkyl chain length. This work studies the effect of chain length to their fluorescent intensity and sensitivity towards HSO3- anion. The type of aggregation-induced emission probes reported here have recently gained attention for applications in sensing and bioimaging.

The authors thoroughly study the binding behaviour of the three fluorescent probes obtained and its aggregation induced emission. Regarding the mechanism for the enhance in emission in the presence of HSO3- the authors provide some evidence based on SEM images, however this is only marginal evidence and it should be mentioned in the main text that the mechanism is assumed. This is an interesting study, and the manuscript is well written, and presented to high standards. This reviewer believes that the readers of Molecules will enjoy the content of the manuscript and considers it suitable for publication after the following minor points have been addressed:

  1. Scheme 1 – The conditions given for obtaining compounds TBPD-10C, TBDP-11C, etc… only mention DMF and 1-Decyl iodide. It should be specified that these conditions only produce TBDP-10C and add the conditions for obtaining TBDP-11C and TBDP-12C.

Response: Thank you for your comments, we have revised these details in Scheme 1 and in the experimental section accordingly. (marked in yellow)

  1. Page 3 – Please report error ranges for the DLS particle sizes. Actually, looking at the DLS distributions for TBPD-10C and TBDP-11C (Figure S13) they are very similar.

Response: Thank you for your comments, we carried out the DLS experiment with a controlling of standard error below 5%, we have added this detail to our manuscript in the experimental section. Though the DLS data of TBPD2+-10C and TBDP2+-11C are very similar, the trend of particle size that positively correlates to the length of alkyl chains can be easily conducted, and such trend becomes clearer after the addition of HSO3-, therefore, we concluded that these probes did combine with HSO3- and further induced a new aggregation form, and the attached alkyl chains contributed to the changes of particle size.

  1. Page 4 final paragraph – Please report the counterions being used to introduce the anionic ions. Do they all share the same cation? Is it possible that there is an influence of the cation?

Response: Thank you for your comments, the series of probes contain two N+ cores, which contributes largely to the ability of anion recognition, and this phenomenon had been reported by many articles. Accordingly, we designed these probes aiming to explore if such sensors could recognize anions, as for the influence of cations, we tested the effects in the competitive experiments (page 5, paragraph 2, co-existence cations contained), the results (figure S17) showed that limited influence on the fluorescent and recognition behavior of such probes had been observed, which as similar as the pre-experiment and corresponds to the theory, so, we did not specially list the cationic ions in page 4.

  1. During the anion binding experiments, a large amount of ions (up to a 100 eq. in some cases) are added to aqueous solutions. Have the authors looked at buffered solutions? Some anions will be more basic than others and the pH and ionic strength of the solutions can have an effect on the aggregation process that the authors might not have considered.

Response: Thank you for your comments, unless otherwise stated, all UV-vis spectra and FL spectra tests were carried out under a Tris-HCl buffer contained condition, and the pH values of the test systems were adjusted to be 7.00 to mimic a more applicable condition for potential use, we now state these experimental conditions in the related captions and in the experimental sections. (marked in blue)

  1. Page 6 – Authors report a DLS measurement as 528.91 nm. This level of accuracy is unrealistic with this technique. Please report a value that takes into account the experimental error of the technique.

Response: Thank you for your comments, the level of accuracy is largely correlated to the dispersion state of testing samples, namely whether a stable correlation function curve could be calculated, once a smooth and ends-near-baseline curve is obtained, we can acquire such submicron data under the condition of standard error below 5%.

  1. Figure 5 – The label for the SEM image in b) is blurry and does not allow to visualize correctly the size reference for the image. Please improve the quality of the text in here to make it easier to read.

Response: Thank you for your comments, we have revised this mistake in our manuscript.

  1. Regarding the aggregation induced mechanism, it would have been interesting to see a thorough pH study of the dependence of fluorescence with pH.

Response: Thank you for your comments, that is a constructive suggestion, we will carry out this suppose in our further research.

  1. Figure S15 – The comparison of the fluorescence spectra does not specify the chain length of TBDP probe (black line). The figure caption does not specify either the number of equivalents of HSO3-for each sample.

Response: Thank you for your comments, the TBDP is the probe without aliphatic chains attaching (Scheme 1), we showed here aiming to illustrate the influence of aliphatic chains on the probe’s recognition behaviors. We have revised our manuscript accordingly.

  1. Figure S16 – Please indicate the magnitude and units for the Y axis in the plots presented.

Response: Thank you for your comments, we have revised this mistake in our manusctipt.

  1. Figure S19 – It is not clear what the authors intend to show with this figure. The authors need to update the caption to explain better what the figure means. Currently it says: “Fluorescent intensity …. Within different pH recognition conditions” What are those conditions? What is the different pH?

Response: Thank you for your comments, we list figure S19 to explain whether different pH values could affect the probe’s recognition and fluorescent behavior, we now revise the caption to “Figure S19. The influence of pH valves on the fluorescent intensity and recognition behavior of TBPD2+-12C and TBPD2+-12C-HSO3- system at 537nm (Taking TBPD2+-12C as an instance, λex=348 nm, voltage: 620V, 5.0 μM sensors, 100.0 equiv. of HSO3-, 2mM Tris-HCl buffer, pH was adjusted by 0.10M HCl and NaOH).” in our manuscript.